# Benefits of Structured Engagement with Target Audiences of a Health Website: Study Design for a Multi-Case Study

**DOI:** 10.3390/healthcare9050600

**Published:** 2021-05-18

**Authors:** Jennifer Tieman, Virginia Lewis

**Affiliations:** 1College of Nursing and Health Sciences, Flinders University, Adelaide 5042, Australia; 2Australian Institute for Primary Care and Ageing (AIPCA), La Trobe University, Melbourne 3083, Australia; V.Lewis@latrobe.edu.au

**Keywords:** palliative care, online health information, knowledge translation, evaluation methods, community engagement, website review

## Abstract

Access to evidence and practice knowledge precedes use, but availability does not guarantee reach and uptake by intended audiences. The CareSearch project provides online palliative care evidence and information to support health and aged care professionals as well as patients, carers and families to make informed decisions about care at the end of life. Already established in the palliative care sector, CareSearch commenced planning to extend its reach, and ensure website use is maximised for different audiences. This paper reports on the development of the Engagement Framework which will be used to guide and deliver an Engagement Project which will actively seek feedback and insights from intended users in a structured process. The process for developing the Engagement Framework commenced with a literature review of approaches used in knowledge translation, implementation science, and social marketing. The Engagement Framework comprising eight steps was then developed. The Engagement Framework outlines the series of tasks to be undertaken by team members when working with three target groups (Aged Care; Allied Health; and Patients, Carers and Families). A process/formative evaluation collecting data using qualitative methods is also described for use in the subsequent Engagement Project. The evaluation will explore the experiences of project participants as well as staff implementing the engagement activities. The three target groups will enable a cross-case comparison of the strengths and weaknesses of the approach. Planning, implementing and evaluating engagement with intended audiences, offers one mechanism to identify ways to increase interaction and integration with knowledge users.

## 1. Introduction

### Background

CareSearch is a nationally established project in Australia providing online evidence and information relevant to palliative care for use by any health professional regardless of setting, and by those affected by the need for palliative care. It aims to support health and aged care professionals, patients, carers and families to make informed decisions about care at the end of life by providing access to trustworthy information and best available evidence. CareSearch is funded by the Australian Government Department of Health [1]. Since its commencement in 2006, the CareSearch Project has evolved and now manages two key websites, the CareSearch site launched in 2008 and the palliAGED site launched in 2017, as well as offering a range of other evidence and practice resources [2].

Palliative care is an area of increasing importance to communities, policy makers and the health system given an ageing population, increasing care complexity relating to disease burden and comorbidities, and the diversity of individual needs of people who are at the end of their life [3,4]. The Australian Government Department of Health has developed a National Palliative Care Strategy and has funded a set of national palliative care projects to improve palliative care awareness and provision across Australia [1,5]. CareSearch is one of these national projects and provides information and resources for both consumers and health professionals regardless of setting of care. CareSearch therefore needs to provide evidence-based information for a diverse set of users and information needs.

To help capture the complexity of its work, particularly for evaluation and strategic development, CareSearch has developed a program logic (Figure 1). In this representation, the CareSearch Project utilises a structured approach to design, development and delivery of content and resources described as the CareSearch Model [6]. This Model includes the functionality within the control of CareSearch, functionality afforded by other systems and platforms, content that is delivered through all systems and methods, and integrated promotion.

Stakeholder engagement has always been a key “input” for CareSearch. In addition to formal governance groups, stakeholders have been engaged to contribute to the quality of content, to improve cohesiveness and reduce duplication in the sector, to contribute to information sharing and skill development across the sector, and to support the promotion of CareSearch resources.

Web metrics reported at the end of 2018 demonstrate significant use, with palliAGED receiving over 10,000 visits monthly and CareSearch receiving over 100,000 visits each month. There are over 8000 people registered to receive newsletters and over 4000 followers of the CareSearch and palliAGED Twitter accounts. The CareSearch blog provides different perspectives on palliative care issues with between 300 and 4300 “reads” per article. Data from multiple sources including newsletter registrations, promotional orders and evaluation surveys all indicate that the resource is being accessed across Australia [1,2].

Evidence is at the core of CareSearch activities [2]. Evidence-based practice respects clinical judgement and the patient’s preferences and wishes while looking to research findings to inform care. It requires health professionals to understand the relationship between research design and strength of evidence, be competent in accessing and appraising evidence, and able to determine its applicability to an individual patient’s circumstances and wishes. Evidence helps determine whether treatments and care practice are effective and of benefit to patients and the health system. Patients, carers and families need evidence-based information to build their understanding of what is happening to them and to support their decision-making. This is particularly important in palliative care where cure is not the measurable outcome.

Having become an established presence in the palliative care sector, CareSearch is seeking to extend its reach, and to ensure that the utility of the website is maximised for all target groups. In an extensive review of evaluations of seven large knowledge mobilisation agencies, Davies et al. [7], noted one of the consistent findings included “the need to tailor research-based resources to the preferences and access needs of different audiences”. User acceptance and confidence in new technologies have been seen as critical aspects of the spread of infrastructure and its subsequent use [8]. Models such as the Technology Acceptance Model [9,10] have looked at the variables that can influence use and usefulness while more recent models such as the NASSS framework have highlighted the need for more nuanced understanding of domains such as the condition or illness, the technology, the value proposition, the adopter system, the organization(s), the wider (institutional and societal) context, and the interaction and mutual adaptation between all these domains over time [11].

A growing literature on the role of website design and user testing has highlighted the importance of design elements such as website architecture, content utility, and readability in supporting website use and engagement [12,13,14,15]. There is also an increasing understanding of the influence of online marketing considerations including search engine optimization [16,17]. However, web elements and acceptance models may not be sufficient to determine the value of an online resource to different potential audiences. Understanding the user and/or the intended audience experience of content and barriers to use and uptake is critical [18]. Previous work had examined the diversity in views and opinions among participants and stakeholders in the development of digital health resources [19]. The influence that online health information has on online health information seekers’ medical decision making is recognised, highlighting the importance of credible online health information [20,21]. Variability in skills to use and access digital health information highlights the need for those providing digital health resources to recognise and respond to this diversity [22]. The importance of understanding context is also argued in implementation science. Lau et al. [23] summarised current knowledge on the barriers and facilitators to the implementation of diverse complex interventions and highlighted the importance of understanding and defining “the context” for individual groups.

Therefore, CareSearch successfully sought funding from the Australian government to design, implement and evaluate a structured engagement activity conducted with three different target groups (Aged care; Allied health; and Patients, carers and families). These three groups represent some of the audiences with which the CareSearch project engages and highlights potential differences in terms of needs, awareness and knowledge of end-user groups. As part of this process, CareSearch will identity issues and considerations around the specific context influencing palliative care awareness and online knowledge use for each target group. CareSearch will use this to inform activities and strategies in the engagement phase to support the utility of the website and, ultimately, uptake of online CareSearch resources by these groups.

The development of a formal Engagement Framework fits into the CareSearch Model at a similar point to other stakeholder activities. The intended consequence of implementing the 2018–2020 structured engagement activity is that CareSearch staff will gain a deeper understanding of the context in which people seek information [24]. These insights will be used to develop a set of priority actions that they would not otherwise have identified that can be implemented to contribute to the functionality, content and promotion of the CareSearch Model in a way that increases its reach and impact with the targeted groups.

The aim of this paper is to report on the development of an evidence-based approach to understanding context to support a targeted engagement approach to influence reach and uptake of online resources. The engagement framework will then be applied in individual studies with each group and these findings separately reported.

## 2. Developing an Engagement Framework

### 2.1. Phase 1. Clarifying the Project Requirements

The initial step in developing the proposed Engagement Framework involved a review of the purpose and requirements of the project as proposed in the grant application. This was undertaken in a detailed meeting between the two authors (the CareSearch Director and project sponsor and an evaluation researcher working with the engagement project). This meeting clarified that the three target groups were fixed and reflected priorities for the project in terms of intended user groups. The Engagement Framework needed to include workshops with sector participants to understand context and determine actions to be implemented by the CareSearch project staff prior to a follow up workshop to evaluate the effect of these actions. This meeting also recognised that the intent of the Engagement Framework was to drive and monitor the engagement activities and hence was a first activity in its own right.

### 2.2. Phase 2: Literature Review to Inform Framework Development

We reviewed literature from a range of disciplines and perspectives to develop the engagement framework and the reporting aspects of the project.

The literature on Knowledge Translation (KT) offers a lens through which to understand the current engagement activity. KT looks at the processes, actors and strategies used to move research and evidence into practice and use to improve outcomes for individuals, systems and societies [25]. A recent scoping review of KT highlighted a range of theories, models and frameworks but reported that there was still little evidence of use in practice [26]. How frameworks can be utilised within the digital health context is also an area of interest [27] as is the need to engage with end-users to develop actionable plans and outcomes [28]. Huzair et al. [29] argue that KT is not just about recognising that stakeholders have “a diverse set of understandings, values and goals, but are influenced by their socio-political and economic environment.” They recommend using a range of inter-disciplinary research methods and approaches to understand better the environment in which knowledge is intended to be used, including undertaking interviews with, and observing a wide range of relevant stakeholders. Jacobson et al. [30] offered a framework “that researchers and other knowledge disseminators who are embarking on knowledge translation can use to increase their familiarity with the intended user groups.”

Processes for engaging stakeholders have also emerged through the practice of KT and are being adapted and adopted more widely. In describing the “Deliberative Dialogues” approach, Lavis et al. [31] describe undertaking stakeholder mapping to generate a list of “decision makers, stakeholders (e.g., professional and civil society leaders) and researchers who will be involved in, or affected by, future decisions related to the issue.” They recommend that participants should be chosen on the basis of explicit criteria, including “the ability…to articulate the views and experiences of a particular constituency on the issue, while constructively engaging at the same time with participants drawn from other constituencies and learning from them” [31].

Based on our review of the literature, we determined that it was important to develop a good initial understanding of the context in which the three target groups were operating prior to bringing people together to talk about the specific issues affecting awareness and use of the CareSearch website. In the absence of this initial step, the process of identifying stakeholders is likely to be limited by the lack of the very context knowledge that is being sought. The purpose of the engagement framework is to go beyond CareSearch’s existing types and levels of stakeholder engagement and extend understanding of context for the three target groups.

### 2.3. Phase 3: Finalising the Proposed Process Based on the Literature Review

Following the literature review, the CareSearch Director and evaluation researcher met again to finalise the Engagement Framework and to formalise a template for documenting information about the group and context based on relevant elements identified in the literature review. The proposed approach was then formally reviewed by the CareSearch and palliAGED Advisory Groups to determine face validity.

Prior to working with stakeholders directly, a systematic analysis of context was undertaken based on public documents and grey literature. Each analysis sought to document as much as possible about the context prior to a face-to-face workshop, and to support identification of potential stakeholders to attend the workshop.

The Engagement Framework includes the following steps:Step 1: Pre-workshop preparation—completion of analysis of context template (see Table 1)Step 2: Organising Workshops
○Step 2a: identifying potential participants○Step 2b: selecting participants○Step 2c: inviting participants○Step 2d: preparing participantsStep 3: Stakeholder Workshop 1: Understanding context and developing potential actionsStep 4: Stakeholder (extended group) webinar: Workshop 1 overview and initial action ideasStep 5: Confirming Action plan and developing operational workplansStep 6: Implementing Action planStep 7: Stakeholder Workshop 2: Action Plan implementation update and assessmentStep 8: Stakeholder Workshop 3: Final (joint) assessment of Engagement Activity

The engagement activity will be implemented by one part-time (0.6FTE) coordinator for each target group and a project manager. The proposed schedule for work is as follows:Engagement Framework Development October 2017–March 2018Engagement Project Establishment: March 2018–June 2018Participant recruitment, workshops, action plan, feedback and reporting: July 2018–June 2020Data analysis: July 2020–December 2020Implication assessment and action planning for CareSearch and palliAGED website and future project communications and end-user engagement: January 2021–December 2021

Figure 2 provides a visual representation of the engagement activity steps. The documentation development process is described below.

### 2.4. Phase 4. Developing the Documentation for the Engagement Activity

To support the activity of the Engagement Project focused documentation is required. The process, purpose and timing of this documentation development is described below.

Step 1: Pre-workshop preparation—completion of template for analysis of context. CareSearch staff will use the template designed for the engagement activity to prepare a document that strengthens and summarises their understanding of the contexts in which palliative care evidence may be accessed and used by the three target groups. The task will be informed by a range of approaches.

The field of implementation science has grown rapidly in response to the recognition that knowledge, skills and behavioural intent are not sufficient in and of themselves to lead to behaviour change. From this perspective, context influences the likelihood that evidence will be used in practice. It has been reported that there are over 60 different models of “knowledge transfer and exchange” in use across the fields of health care, social care, and management [32], and a number of attempts have been made to map these in a way that can support knowledge translation practice and evaluation [7,33,34]. Most of these approaches recognise that context is important and suggest general broad categories to be considered when implementing a practice intervention. Lau et al. [23] describe four domains relevant to introducing practice change in primary care organisations; each includes a large number of potential areas to consider.

External—including: the presence and nature of national and local policies and legislative or regulatory mechanisms; incentives and penalising consequences; commonly held values and beliefs; buy-in by stakeholders at different levels; infrastructure; technology; economic climate and financing models; and public awareness.Organisation—including: culture and leadership, which contribute to organisational readiness; resources including time, funding, staff, and technical support; processes and systems; relationships between workers and patients; skill mix and clarity of roles and responsibilities; and team involvement including collaborative working and shared vision.Professionals—including: professionalism including skills and self-efficacy; underlying philosophy of care; attitudes, motivation, and experience; and competencies, including those acquired through training.Intervention—including: the nature and characteristics of the “intervention” including its complexity, evidence of its benefits, cost-effectiveness, and compatibility to the setting; implementability; and safety and data privacy concerns being addressed.

Jacobson et al. [30] offered a framework “that researchers and other knowledge disseminators who are embarking on knowledge translation can use to increase their familiarity with the intended user groups”. Their framework includes questions that can help to generate and organise information in four areas: the user group, the issue, the research, the researcher-user relationship, and dissemination strategies [30].

Social marketing approaches to analysing situation and influencing factors offer another perspective for understanding context. Used in public health, “social marketing approaches can be used to help engage end-users in the development, implementation and evaluation of policies and programmes” [35]. Within a social marketing approach, a situation analysis helps to identify key issues that may impact on a programme or activity or on “the receptivity of target audiences” [35]. There are several tools that come from business and marketing paradigms, that can be helpful when adopting a social marketing approach for public health, such as a PESTLE (Political, Environmental, Social, Technological, Legal and Ethical issues) analysis [36].

All of the above approaches offer some guidance to the task of documenting the context in which the three target groups access and use evidence and resources, but none is perfectly aligned. Therefore, we adapted and combined relevant elements to design a template that CareSearch staff will use to develop an initial description of context for each target group. While the template reflects other approaches, it does not map directly to them. However, the template recognises the different spheres of influence on behaviour described by Lau [23], Jacobson [30] and others [29,32,33,34]. Section B4 addresses the range of context issues in five parts from the broad socio-political (Section B1) to settings including organisations (Section B2) and finally individuals (Section B3). In this case, individual target group members include community members as well as health professionals. Within these broad headings, questions include themes reflected in PESTLE, covering policies, technology, and communication mechanisms. The potential role of key influencers that is a key consideration in social marketing approaches is also captured. Sections B4 and B5 reflect some of the key constructs from implementation science such as the nature of “the intervention” from Lau [23] and “the issue” from Jacobson [30], and the way evidence is recognised and adopted.

The template will also address issues around the online aspect and its relationship to the user group. Some of the elements that are used to describe context for the Patients, Families and Carers (PFC) target group will be different to those used to describe the aged care and allied health groups. There are no pre-defined boundaries to the PFC target group, although some may be developed for the purpose of the engagement activity. For example, the PFC target group may focus on people who have current or recent experience of palliative care and/or end of life.

Step 2: Organising Workshops
Step 2a: Identifying potential participants: Potential participants will be identified for each of the three target groups through targeted invitations to individuals or organisations identified in the context document and through a general call for interested people made on the CareSearch website. All potential participants identified will be documented. (See Table 2)Step 2b: Selecting participants: The CareSearch team will assess each potential participant’s inclusion, indicating “yes”, “maybe”, or “no”. Ratings will be compared at a team meeting and a final set of 20–25 participants will be invited with the aim of having 15–20 participants attending Workshop 1.Step 2c: Inviting participants: The coordinator for each target group will contact potential participants by telephone or email to invite them to participate. In some instances, the coordinator will need to explain why the person has been invited, as they may not see the immediate relevance in attending a workshop organised by CareSearch around palliative care knowledge.Step 2d: Preparing participants: Stakeholders who agree to participate will be provided with a range of CareSearch/Palliative Care in Aged Care Evidence (PCACE) resources intended for use within their target group. Each group will be provided with a sector specific resources pack. This will be a targeted summary of online resources perceived to be relevant from within CareSearch including webpages, evidence resources, learning modules, videos, examples of case studies etc. Participants will be asked to familiarise themselves with the web resources and website prior to attending a workshop. They will also have further information about what to expect at the workshop, including what the boundaries of disclosure of information that is shared will be.

Step 3: Stakeholder Workshop 1: Understanding context and developing potential actions. Through a facilitated process, a workshop will be conducted with each target group separately during which participants will be asked to validate (or challenge) the perceptions that CareSearch has about context and identify gaps in the description of barriers and facilitators to reaching the target groups and encouraging them to access and use PC evidence and prioritise potential areas for action. While similar domains will be explored with all groups, the questions used to generate discussion and descriptions will differ for the health professional groups and the patient, family and carer group. Following the workshops, notes will be collated, and ideas generated will be analysed to identify themes for areas of action to be included in draft Action Plans.

Step 4: Stakeholder (extended group) Webinar: Workshop 1 overview and presentation of initial action ideas. A webinar will be held with each group and all potential participants will be invited to attend. The focus of the webinar will be on presenting a summary of discussions and ideas generated at the workshop, and inviting participants to comment, including challenging or adding to actions and strategies. Inviting all potential participants extends the reach of the engagement activity and provides some continuity of participation if future workshops have space available for new invitees because past attendees cannot attend. It is anticipated that between 5–20 participants will attend the Stakeholder Webinar.

Step 5: Confirming Action Plan: CareSearch staff involved in the Engagement. The Action Plans will be finalised using structured processes and discussion, including assessment of the feasibility of implementing actions within the 12-month time-frame of the project. It is expected that there will be common areas of action. The final action plans will also identify indicators that can be used to assess the impacts of the action plans. Examples of possible measures include webmetric tracking, website pattern usage, impact on digital referrals or measures of reach and uptake in target audiences.

Step 6: Implementing Action Plan: CareSearch staff will implement the Action Plans. The evaluation plan will be developed further based on the contents of the Action Plans.

Step 7: Stakeholder Workshop 2: Action Plan implementation update and assessment. Each group of stakeholders will be invited to attend a second workshop during which the status of implementation of the Action Plan will be reviewed and the findings of the evaluation will be presented. Stakeholder participants will be asked for their reflections on the impacts of the Action Plan, and their perceptions of the extent to which the Action Plan implemented reflected the original workshop.

Step 8: Stakeholder Workshop 3: Final (joint) assessment. Representatives from the three focus groups will come together to attend a final workshop during which they will be invited to reflect on the process and their participation in the stakeholder engagement activity.

### 2.5. Phase 5. Refining Evaluation Design

The engagement framework is intended to provide a structured approach to understanding the context that affects the reach and uptake of evidence in each target group to inform development of clear priority actions that CareSearch can implement and evaluate. The overall goal of implementing and evaluating the engagement activity is to consider whether this approach adds value to the work of CareSearch (formative or process evaluation focus), and whether it increases reach and utility of the website and uptake of resources to support implementation of evidence in the target groups (summative or impact evaluation focus).

The key evaluation questions are:Is the structured and consistent approach to understanding context that was implemented (through structured desk-top review and stakeholder workshops) an effective way to become more familiar with target groups?
What differences, if any, were there in the effectiveness?How do participants experience the workshops? What worked well; what could be improved?Do the workshops lead to Action Plans that include novel prioritised actions and strategies that CareSearch can implement?Do the actions and strategies developed through the engagement activity achieve their intended impacts?What is the overall assessment by key stakeholders of cost-benefit?

## 3. Proposed Methods

With the Engagement Framework described, it was then possible to outline data elements and data collection methods that will be used to capture aspects of the Engagement Project activity. These will form the results and findings of the Engagement Project as it is undertaken. Data collection activities will include:Individual participant surveys following Workshops 1 and 2 with a focus on assessing:
○Prior knowledge (use, etc.) of CareSearch○Prior knowledge of palliative care, experiences○Expectations of engagement activity○Experience of participationSemi-structured interviews and/or group discussions—to be conducted with stakeholders at multiple time points, including:
○CareSearch team members as part of regular meetings to monitor the process, document decisions, and identify issues as they arise○CareSearch team members individually after Workshops 1 and 2, and at the end of the project, to capture key experiences around implementation of the engagement activity○Workshop participants at the completion of the engagement activity—through facilitated discussions at Workshop 3Review of documents used to conduct the engagement activity, including:
○Context analysis reports○Lists of potential workshop participants○Workshop notes○Action Plans○Documents developed as part of Action PlansAssessment of Routine Data collected by CareSearch, including Webmetrics. There may be potential to compare trends for web contact associated with the three target groups and those target groups that are not the focus of the engagement activity; however, note that the three target groups were chosen because of the potential challenges of reach and implementation, so the potential to attribute change or differences to the engagement activity may be limited.

## 4. Results Framework

Ethics approval for the evaluation has been given by La Trobe University Science, Health and Engineering College Human Ethics Sub-Committee (HEC18116).

Key steps of the engagement activity framework have been implemented and collection and analysis of data is currently underway.

## 5. Discussion

There is a lot of emphasis on stakeholder engagement in policy and practice, but little evidence to demonstrate that a systematic approach to understanding context based on literature review, mapping and stakeholder workshops can lead to novel actions [37,38]. Despite the enthusiastic adoption of concepts such as stakeholder engagement and “co-design”, there is very little evidence to support particular actions that will lead to impacts that could not have otherwise been achieved [39]. This study attempts to address this issue in an online context.

More importantly this paper outlines a framework to investigate how context and target group characteristics can reflect barriers and facilitators to dissemination, participation and utilisation of online health information. There is increasing use of online health information by health consumers and health professionals in medical and care decision making. Understanding how to build awareness of the value and relevance of online evidence to discrete groups or sectors and how to identify and measure change in reach and uptake of such evidence, could be instrumental in improving the use and value of evidence-based practice within the health system and the community. The results will make an important contribution to the evidence base for digital translation of evidence and inform our understanding of factors, contexts, and characteristics influencing evidence use.

The creation of context prior to the workshops will enable an assessment as to whether a preliminary review is sufficient to develop an effective and useful understanding of the context for online information use, or whether direct face-to-face engagement with the studied group itself reveals additional, and critical, information. By simultaneously studying the process and outcomes of engagement with three discrete groups we will be able to explore what is common and what is unique about the nature of digital audiences and whether action strategies need to be target group specific or are universal in nature. It is also anticipated that the study will provide valuable insights into the contribution of marketing and communication strategies to digital translation.

The structured engagement framework and associated evaluation provide rich opportunities to assess not only the influence of the characteristics of the target groups but also the influence of staff involved in the project on the engagement process. The evaluation will consider whether and how a focus on engagement influences the delivery of online health information. This is particularly critical for virtual projects where opportunities for incidental interaction with intended audiences may be more limited [24].

## 6. Limitations

There is a rich and varied literature relating to KT, digital health processes and consumer/end-user engagement. Our literature review was selective and may have missed relevant materials. The development of the engagement framework was time sensitive as it was a precursor to the dependent activity of participant selection, workshops and action plans. Hence there may be elements that could inform the framework that have been missed. The framework depends on engagement from the targeted audiences and it is not known if there will be sufficient interest in participation. The framework assumes that it will be possible to identify common and unique elements from the different audiences that could assist in developing specific content and/or communications strategies.

It should also be noted that the framework was designed for a single website with the specific health focus of palliative care. This will limit unthinking generalization of the framework. However, the CareSearch Project is designed to serve a broader audience group, form those affected by the need for palliative care, to those providing palliative care in any health care setting.

Despite these limitations, there is both a practical and a theoretical need to understand more about how different users engage with and respond to digital content and to the mechanisms by which such content is promoted and distributed.

## 7. Conclusions

While evidence retrieval and synthesis activities are critical to the development of trustworthy online health information, we also need to address how to support the use of evidence in practice. A structured approach not only enhances the value of our interaction with intended users but provides us with the ability to evaluate the effectiveness of the framework approach and its contribution to online knowledge translation.

## Figures and Tables

**Figure 1 healthcare-09-00600-f001:**
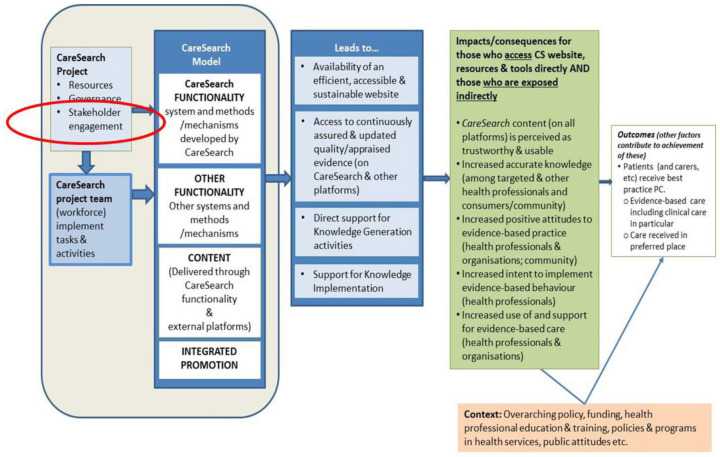
CareSearch Evaluation Program Logic (Stakeholder engagement highlighted).

**Figure 2 healthcare-09-00600-f002:**
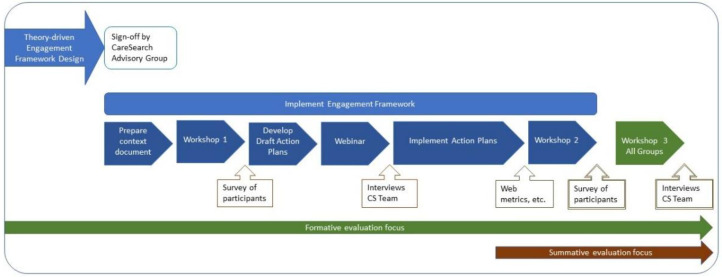
Timeline for Engagement Activity Steps and proposed data collection.

**Table 1 healthcare-09-00600-t001:** Template for describing context systematically across target groups.

Introduction
A standard introduction describes the aims of the context mapping exercise. The person completing the template describes the method they used.
**Part A**
In Part A there are five core questions to consider in broad terms:
What are the characteristics of the target groups? In what settings are the target groups found? How can they be reached? What contextual factors affect the way the target group accesses and uses information and evidence in general? What role do the target group have in palliative care? What information do they need? Where do they need it? How do the target group currently engage with CareSearch, including using evidence and resources?
**Part B**
In Part B more specific questions are addressed for context issues related to policy, training, organisational networks, and sector understanding and use of evidence in palliative care.
***Section B1—Broad Socio-Political Context***
Document what is known about:
• Policies and legislation that affect the target group, including economic climate and governmental financing. ○ Consider policy and regulations (governmental or other central entity), external mandates, recommendations and guidelines, pay-for-performance, and public or benchmark reporting that affect behaviour in the target group. ○ What incentives exist for the user group to use research/evidence? • Who are key influencers in the sector? ○ Identify individuals and groups who influence attitudes, knowledge, and behaviours of the target group, who might be involved through strategies such as social marketing, education, role modelling, training, and other similar activities. Including: opinion leaders formally appointed internal implementation leaders champions external change agents • What is the nature and quality of links and social networks and the nature and quality of formal and informal communications across the target group? ○ Consider the extent to which organisations and groups are linked and networked. • What kind of technology is routinely available and used by the target group? ○ How would technology affect the target group’s access to information/evidence? • What is the level of awareness of palliative care in the target group? ○ Are there sub-groups with good awareness? What characterises them?
***Section B2—Context in Settings in Which Target Group Members Are Present***
• What kinds of organisations and settings do members of the target group operate in and with? ○ In what formal or informal structures is the user group embedded? To whom is the user group accountable? ○ Consider workplaces, networks, professional associations, etc. • What level of support for use of palliative care evidence is there among the organisations, networks, etc? ○ Consider whether there is peer support across organisations or competition. • What is the typical nature and quality of formal and informal communications in organisations? • What is the level of readiness to support activities that would seek to promote access to and uptake of palliative care resources? ○ Describe any evidence that the organisations/settings would be committed to supporting strategies • What is the skill mix across the organisations where the target group is found? ○ Consider whether the skill mix affects the way the target group would access or use evidence.
***Section B3—Characteristics of Individual Target Group Members***
• What is the scope of practice and the professional roles of target group members? ○ What criteria do the user group use to make decisions? ○ Consider how professional roles would support or undermine access to and use of Palliative Care (PC) evidence • What philosophies of care operate within the target group? ○ Consider whether there is a general philosophy or whether there are differences between sub-groups. • How open to change in general are members of the professional? ○ Consider whether there is evidence that the target group is • What competencies to the target group members have that are relevant to accessing and using PC evidence?
***Section B4—about Evidence Awareness and Use by the Target Group***
• For what purposes does the user group use evidence? ○ How sophisticated is the user group’s knowledge of research methods and terminology? • Where do the target group generally go to find good quality evidence? ○ What sources of information does the user group access and use? ○ How does the user group process information—i.e., how does it access, disseminate and apply information? • What arguments would convince the target group of the need to prioritise use of PC evidence? • Do the target groups perceive CareSearch to be well presented? • Are there any costs associated with accessing and using PC evidence?
***Section B5—about current approaches to implementing innovations in the sector***
• What evidence is there that the target group/sector has capacity to plan and implement interventions designed to support practice change? • Consider whether the target group has experience implementing practice change interventions. (What capacity building might be helpful?)
**Part C**
In Part C, the person completing the context mapping document is asked to reflect on the process by considering three questions:
Did you find the table useful in helping you to answer the five broad questions about context? If not why?/ If yes how? (in broad terms) Do you think your understanding of the sector improved after completing the context document? Are there any questions/topics that you would add for the sector you researched? A final section is provided for “Other questions and comments”.

**Table 2 healthcare-09-00600-t002:** Table to document identification of stakeholders.

Name of Person, Organisations, Role, or Group to Be Considered	Source of Nomination—Category (e.g., Social Media, CareSearch Staff Member or Other Person Recommending, etc.)	Why They Have Been Nominated or Are Self-Nominating. (What They Might Bring to the Task.)	Name of Person Nominating (Where Relevant)

## Data Availability

Not applicable.

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
