# Peer review of "Benefits of Structured Engagement with Target Audiences of a Health Website: Study Design for a Multi-Case Study"

_healthcare, 2021, doi:10.3390/healthcare9050600_

Round 1
Reviewer 1 Report
Thank you for the opportunity to review this article. The information presented is an important topic and worthy of scientific exploration. As we move into an era of increased technological use, the methods, evaluation, and reporting of the scientific information regarding the use of technological modalities and specific tech information will be important. This information will be important as patients, care providers, and healthcare workers all extend their use of various forms of technology, hence, the protocols to drive these expolorations are also important and deem significant consideration. This article presents a specific protocol for use of a website for patients and caretakers. It is important to optimize this use and present worthwhile information for these important stakeholders.
First, I believe the authors have done a good job presenting their case and their specific development of their protocols, methods, and underlying theory driving their upcoming project. Considerable thought was presented in the development of these protocls and this is evident in the literature search, logical flow of the protocols, the methods, and the upcoming project implementation. The references, synthesis of the information, and the utlization of the materials all seem appropriate.
There seems to be a missing important reference, that at least deems refernce to and some discussion. Venkatesh & Bala (2008) with their TAM 3 model (see reference below) have a similar widely used model for technology and acceptance. My question, was this considered in the development of the protocols and the literature search? It seems that the protocols could have been developed with an already established theoretical model that has already been validated and revised based on the prior literature associated with the TAM, and now already with the TAM 3. Can the authors make comment on this? And I would recommend some comment in the literature why this was not used or why this was avoided. Overall, I think some comment on the TAM 3, either a justifcation for not using or an integration into the given literature, would help strengthen this article and the overall project.
Second, while reading the manuscript I kept coming to the conclusion that this was a very specific and tweeked protocol, based on the given literature, for the specific situation of the CareSearch website. From this, I kept thinking that this protocol was so specific to the one project the generalizability of the project will need to be reviewed. Therefore, the question is how can this be used for other projects, situations, environments, etc. Some information of this would be important in the discussion, limitations, and possibly even the introduction. There was some reference to the importance of protocols for technology driven implementation and research, but it was left at that, small reference.
Finally, overall, I think this is an important subject and worthwhile manuscript. With some minor editing and additions, I believe this article will be rigorous and viable. Thank you for the opportunity to review this manuscript and I wish the author team much success.
Venkatesh, V. and H. Bala (2008). "Technology Acceptance Model 3 and a Research Agenda on Interventions." Decision Sciences 39(2): 273-315.
Reviewer 2 Report
- The "Introduction" section has to be completed with references related to mediatization theories :
French Media Representations towards Sustainability: Education and Information through Mythical-Religious References, Sustainability, 12(5) : 1-18, 2095; https://doi.org/10.3390/su12052095 - The "Results" has to be completed.
- The "Discussion" has to be completed with elements that take into account the relationship between Aging and Digital Communication. References available here: https://www.essachess.com/index.php/jcs/issue/view/27
- Aspects relating to the mediatization of health must be integrated in the "Conclusion": elements available here: https://www.mdpi.com/1660-4601/18/5/2287
Reviewer 3 Report
The authors have to address all comments/concerns carefully to ensure that all issues affecting the sobriety of the study are eliminated.
- Authors should improve the article title to better describe the topic.
- The abstract requires improvement because it is not very clear. The authors should explain the study problem, methods of study, and then findings of the study.
- Introduction Section: The introduction requires adding references, some sentences and paragraphs do not contain references. Who is meant by "they" (page3-line77)? The introduction contains many unconnected sentences that make the introduction unclear and targeted. It is preferable to add the contributions of this study at the end of the introduction in bullet points (with exact description). The introduction is very long and contains many sub-addresses. The introduction section requires improvement.
- In subsection “Literature Review to Inform Framework Development”, some information should be transferred to a separate subsection such as (pages4-5, lines171-196).
- Is the sentence "Developing the Documentation for the Engagement Activity" the title of the subsection (Page5-line199)?
- What are “Section B1”, “Section B2”, “Section B3”, … etc. (page6)? Authors should describe these sections so they are clear to the reader before using them in Table 1.
- The methodology section is not properly described.
- Why the authors focused on the health care website (CareSearch) and not others Although there are many health care websites/projects in Australia. The authors should explain the advantages of building a framework with CarSearch compared to other projects.
- This study protocol requires re-organization and restructuring.
- Figures: Figures 1 and 2 are unclear and blurry, authors should redraw these figures with high resolution. Figure 2 is not invoked in-text.
- There are many sentences that require paraphrasing in different places of the article, the authors must scan the entire article to find these sentences and rewrite them.
- List of references: The references require little improvement. For instance, the number of references in this article is 24, and the number of old references is 14. Therefore, we recommend updating the list of references with some related/recent research. Some search names in the references list begin with an uppercase letter in each word such as [2] and [3], and other words begin with a lowercase letter, such as [1] and [6]. Authors should standardize the writing style of research names. Is reference [21] necessary? It seems not quite relevant to the topic of the study. This study needs a minor check of the reference list.
- English writing: This study needs proofreading of the entirety of the research to remove all the issues related to typos, spelling, and grammar mistakes. For instance, some of the words need "a", "an" and "the" in different places in this article (such as “promotion” (page2-line51), “clinical”, (page2-line61), … etc.), replace “Evidence based” with “Evidence-based” (page2-line61), add space between “engagement” and “[“ (page3-line77, check the entire article), replace “end user” with “end-user” (page3-line97), remove space between “)” and “.” (page3-line120), use “have” instead of “has” (page6-line255), replace “to” with “from” (page6-line263), … etc. Authors must remove all grammatical and typo problems to improve the quality of English writing and to make this study mistakes-free.
Round 2
Reviewer 2 Report
Very insufficient scientific framework for publication.
The readers cannot objectively make sense of the results presented.
Author Response
Reviewer 2
Very insufficient scientific framework for publication.
We do not accept this statement from Reviewer 2. The article presents a clear statement that the purpose of this article is to report on the development of an Engagement Framework for use in a subsequent activity, the Engagement Project. The paper reports on the review of the literature and evidence base that underpinned the structure of the Engagement Framework and details the elements and steps of the Engagement Framework that will be used in conducting the Engagement Project. In essence, it describes the protocol or methods for the actual study which will be reported later. We have revised the Abstract to try and clarify the process of the development of the Engagement Framework that will underpin the actual Engagement Project.
The readers cannot objectively make sense of the results presented.
The “results” reported in this article are the description of the Engagement Framework to be used in the subsequent Engagement Project. As stated in the Abstract and in the Introduction and the aim, the purpose of this article is to outline the details of the Engagement Framework which is the description of how the Engagement Project will be conducted. The results of the Engagement Project itself will be reported separately. To try to clarify this for the reader we have added the following sentence to the Proposed Methods section
With the Engagement Framework described, it was possible to outline data elements and data collection methods that would be used to capture aspects of the Engagement Project activity. These will form the results and findings of the Engagement Project as it is undertaken.
Reviewer 3 Report
The authors did not respond to some comments carefully.
- The scientific abstract should be clear, comprehensive, and independent for the readers. The authors should first describe the purpose of this study. The methods are described by the authors and the findings are not necessarily "results" (we did not mention the word “results” at all). Findings describe what is extracted from the study (whether study protocol, concept paper, review, ... etc.). The abstract still requires improvement. The authors did not address this comment appropriately.
- Introduction Section: The authors did not add references to some paragraphs (the first five paragraphs), for example, how the authors knew "the CareSearch site launched in 2008 and the palliAGED site launched in 2017", "CareSearch receiving over 100,000 visits each month.", ... etc.
- In subsection “Literature Review to Inform Framework Development”, some information should be transferred to a separate subsection such as (pages4-5, lines171-196). The authors did not address this comment appropriately.
- The methodology section is not properly described.
- Why the authors focused on the health care website (CareSearch) and not others Although there are many health care websites/projects in Australia. The authors should explain the advantages of building a framework with CarSearch compared to other projects. The answer should be added to the Introduction or Literary Reviews section and not the limitations section. The authors pointed out “this study has been undertaken as part of the CareSearch Project”, this is not answer. Authors should add a paragraph explaining what distinguishes CareSearch from others.
- Figures: Figures 1 and 2 are unclear and blurry, authors should redraw these figures with high resolution. The authors did not address this comment appropriately.
- Paraphrasing: There are many sentences taken from previous research. Some of this research is included in the list of references, others are not. Authors must paraphrase all sentences taken from the previous research. The authors must avoid taking whole sentences from the original papers (even if these papers/articles/websites were their previous research). They are obligated to paraphrase during the revision of the entire article. Also, authors should include missing references. For instance, “This is particularly critical for virtual projects where opportunities for incidental interaction with intended audiences may be more limited.”, …etc. Please check the entire paper.
- English writing: This study still requires minor proofreading of the entirety of the research to remove all the issues related to typos, spelling, and grammar mistakes.
Author Response
The scientific abstract should be clear, comprehensive, and independent for the readers. The authors should first describe the purpose of this study. The methods are described by the authors and the findings are not necessarily "results" (we did not mention the word “results” at all). Findings describe what is extracted from the study (whether study protocol, concept paper, review, ... etc.). The abstract still requires improvement. The authors did not address this comment appropriately.
As noted in the abstract the purpose of this study is to “report on the development of the Engagement Framework to be used to guide and deliver the Engagement Project”. This paper does not report “results”. The “results” reported in this article are the description of the Engagement Framework to be used in the subsequent Engagement Project. As stated in the Abstract and in the Introduction and the aim, the purpose of this article is to outline the details of the Engagement Framework which is the description of how the Engagement Project will be conducted. The results of the Engagement Project itself will be reported separately. We have made changes to the Abstract to try and clarify the purpose and process taken in developing the Engagement Framework.
Introduction Section: The authors did not add references to some paragraphs (the first five paragraphs), for example, how the authors knew "the CareSearch site launched in 2008 and the palliAGED site launched in 2017", "CareSearch receiving over 100,000 visits each month.", ... etc.
We thank the reviewer for this comment. We have added the following references that address this issue.
Tieman, J. Ensuring Quality in Online Palliative Care Resources. Cancers 2016, 8, 113. https://doi.org/10.3390/cancers8120113
URBIS Evaluation of the National Palliative Care Projects: Final Report 2016 Available online at https://www.health.gov.au/resources/publications/evaluation-of-the-national-palliative-care-projects-final-report (accessed 30 April 2021)
In subsection “Literature Review to Inform Framework Development”, some information should be transferred to a separate subsection such as (pages4-5, lines171-196). The authors did not address this comment appropriately.
We have not moved the information but have instead created an additional phase (Phase 3: Finalising the proposed process based on the literature review) which clarifies the stage where the knowledge from the literature was used to develop the actual Engagement Framework.
The methodology section is not properly described.
We have added the following sentence to the Proposed Methods
With the Engagement Framework described, it was possible to outline data elements and data collection methods that would be used to capture aspects of the Engagement Project activity. These will form the results and findings of the Engagement Project as it is undertaken.
Why the authors focused on the health care website (CareSearch) and not others Although there are many health care websites/projects in Australia. The authors should explain the advantages of building a framework with CareSearch compared to other projects. The answer should be added to the Introduction or Literary Reviews section and not the limitations section. The authors pointed out “this study has been undertaken as part of the CareSearch Project”, this is not answer. Authors should add a paragraph explaining what distinguishes CareSearch from others.
We have added the following extra paragraph with references in the Introduction. It contextualises the role of CareSearch in palliative care. This is simply a statement as to the role of CareSearch as we would not have undertaken this project for any other website. It is intrinsically linked with the role of CareSearch.
Palliative care is an area of increasing importance to communities, policy makers and the health system given an ageing population, increasing care complexity relating to disease burden and comorbidities, and the diversity of individual needs of people who are at the end of their life [34, 35]. The Australian Government Department of Health has developed a National Palliative Care Strategy and has funded a set of national palliative care projects to improve palliative care awareness and provision across Australia [31,32]. CareSearch is one of these national projects and provides information and resources for both consumers and health professionals regardless of setting of care. CareSearch therefore needs to provide evidence-based information for a diverse set of users and information needs.
Figures: Figures 1 and 2 are unclear and blurry, authors should redraw these figures with high resolution. The authors did not address this comment appropriately.
We previously submitted a high-resolution version of this figure to the journal.
Paraphrasing: There are many sentences taken from previous research. Some of this research is included in the list of references, others are not. Authors must paraphrase all sentences taken from the previous research. The authors must avoid taking whole sentences from the original papers (even if these papers/articles/websites were their previous research). They are obligated to paraphrase during the revision of the entire article. Also, authors should include missing references. For instance, “This is particularly critical for virtual projects where opportunities for incidental interaction with intended audiences may be more limited.”, …etc. Please check the entire paper.
We have checked the paper. Additional references have been added.
English writing: This study still requires minor proofreading of the entirety of the research to remove all the issues related to typos, spelling, and grammar mistakes.
The paper had been reviewed and proofread prior to resubmission by another member in our research group.